# Association between Dietary Practice and Gait Speed in Community-Dwelling Older Adults with Overweight and Obesity: A Cross-Sectional Study

**DOI:** 10.3390/diseases12030054

**Published:** 2024-03-08

**Authors:** Leila Itani, Massimo Pellegrini, Dana Saadeddine, Hanen Samouda, Dima Kreidieh, Hana Tannir, Marwan El Ghoch

**Affiliations:** 1Department of Nutrition and Dietetics, Faculty of Health Sciences, Beirut Arab University, Riad El Solh, P.O. Box 11-5020, Beirut 11072809, Lebanon; l.itani@bau.edu.lb (L.I.); dana.saadeddine@gmail.com (D.S.); d.kraydeyeh@bau.edu.lb (D.K.); hana.tannir@bau.edu.lb (H.T.); 2Center for the Study of Metabolism, Body Composition and Lifestyle, Department of Biomedical, Metabolic and Neural Sciences, University of Modena and Reggio Emilia, 41125 Modena, Italy; massimop@unimore.it; 3Department of Precision Health, Nutrition and Health Research Group, Luxembourg Institute of Health, 1A-B, Rue Thomas Edison, Strassen, L-1445 Luxembourg, Luxembourg; hanene.samouda@gmail.com

**Keywords:** BIA, nutritional status, diet, overweight, obesity, physical performance, gait speed

## Abstract

Slow gait speed is associated with poorer clinical outcomes and higher rates of functional limitation and mortality in older adults, especially when combined with overweight or obesity. Aging is also associated with nutritional deficits. The aim of our study was to assess the potential association between dietary practice and gait speed performance in community-dwelling older adults with overweight and obesity. Participants underwent body composition measurement with the Tanita MC-780MA Bioimpedance Analyzer (BIA). Dietary patterns were assessed with the Mini Nutritional Assessment (MNA) questionnaire, and a dietary adequacy (DA) score system was constructed. The four-meter gait speed test was performed in order to assess gait speed. Of 222 participants, aged 67.6 ± 6.6 years, with a body mass index (BMI) of 31.9 ± 4.5 kg/m^2^, 34.7% had reduced gait speed and lower DA compared to those with normal gait speed (2.99 ± 1.12 vs. 3.37 ± 1.07; *p* < 0.05). The DA score of participants with slower gait speed was more likely to fall below the median than that of participants with normal gait speed (70.1% vs. 51.7%; *p* < 0.05). Participants with slower gait speed were more likely to be nutritionally at risk of low DA (22.1% vs. 10.3%; *p* < 0.05). Logistic regression analysis, after adjustment for confounders, showed that the risk of having a slow gait speed was 75% lower among those with a higher DA score (OR = 0.25; 95% CI = 0.11–0.53). Older adults with overweight or obesity in community dwellings might need to be supported with nutritional interventions that can improve their gait speed.

## 1. Introduction

Older adults have a higher risk of age-related decline in functional performance [1], associated with physical health-related quality of life (HRQoL), falls, health care use, and admission to residential care [2,3]. One of the most widely used tests to assess functional performance [4], is the four-meter gait-speed test, since it is reliable, easy and quick to perform, and inexpensive [5,6]. It is defined as the time an individual takes to walk a specified distance, on level surfaces, over a short distance [7]. It has been considered by the International Academy on Nutrition and Aging Task Force as the “sixth vital sign”, since it is a health indicator during aging [8] and is able to predict health-related outcomes, e.g., clinical (geriatric syndromes and rate of hospitalization), cognitive (Alzheimer disease and dementia), functional (falls and physical activity), HRQoL, and survival and mortality rates [9,10,11,12,13,14,15,16,17]. For this reason, over the last decade, gait speed tests have gained interest in clinical research settings, and identifying strategies to improve gait speed is seen as extremely important.

Another growing public health challenge among older adults is overweight and obesity, which constitute chronic relapsing diseases [18] and have been demonstrated to further increase the risk of functional impairment and limitations related to aging [19].

In fact, in terms of older adults, overweight or obesity specifically has been shown to negatively impact gait speed, leading to functional impairment [20]. In addition, nutritional inadequacy (i.e., under-/over-nutrition) is highly prevalent among older people and could be exacerbated by overweight or obesity [21]. Thus, dietary adequacy (DA), defined as the sufficient dietary intake needed to fulfill nutritional requirements for optimal health, needs to be carefully monitored and managed in older people who are overweight or obese, since according to the defined criterion of adequacy, the requirement for a given nutrient may be at a lower or higher intake amount [22].

The recent literature has also highlighted an association between slow gait speed, overweight, and obesity in a nationwide population-based sample of older adults [23]. Indeed, in this study, an association between nutritional status and gait speed was found and reported [23]. However, despite this interesting finding having not been widely examined by many studies [23], it remains not fully understood. Based on these considerations, the aim of the present study is to assess overall gait speed, the prevalence of low gait speed among community-dwelling older adults with overweight and obesity, and its association with dietary practice. We expect to find an association between low DA and slow gait speed.

## 2. Materials and Methods

### 2.1. Participants and Design of this Study

This study is a cross-sectional study conducted in the Department of Nutrition and Dietetics at Beirut Arab University (BAU), Lebanon, between March 2018 and February 2020 [24]. A total of 222 participants were included. Participants were recruited from the general population through an e-mail-based survey targeting self-sufficient, independent, community-dwelling older adults, including those employed at BAU, aged 60 years and older and with a Body Mass Index (BMI) ≥ 25.0 kg/m^2^. The included participants were those classified as overweight (BMI ≥ 25.0 kg/m^2^) or obese (BMI ≥ 30.0 kg/m^2^) based on the National Institutes for Health (NIH)/World Health Organization (WHO) guidelines for BMI classification. The exclusion criteria were: (i) aged under 60 years; (ii) not living independently and living in settings other than the community; (iii) underweight (BMI < 18.5 kg/m^2^) or normal weight (18.5 kg/m^2^ ≥ BMI < 25.0 kg/m^2^) based on NIH/WHO guidelines; (iv) unable to walk without the help of assistive devices (i.e., crutches, walker, etc.); (v) having artificial limbs or limb prostheses; and (vi) have been currently diagnosed with cancer, cirrhosis, or chronic organ failure (i.e., renal, heart, liver, etc.).

This study has been approved by the Institutional Review Board of the Beirut Arab University (IRB-BAU) (No. 2019H-0063-HS-M-0318). All those who participated in this study gave informed and written consent for the use of their personal data in an anonymized way. An internal standard record of our department was administered to all participants with information regarding their medical history, lifestyle habits, and demographic and social conditions.

### 2.2. Body Weight and Height

Body weight and height were determined by means of an electronic weighing scale (SECA 2730-ASTRA, Hamburg, Germany) and stadiometer, with individuals wearing light clothes and no shoes. The BMI was calculated by dividing bodyweight in kg by the square of the height in meters according to the standard formula:Body weight (kg) ÷ height^2^ (m).

### 2.3. Body Composition

A multi-frequency Bioelectrical Impedance Analyzer (BIA) (MC-780MA, Tanita Corp., Tokyo, Japan) was used to measure performance in the morning by the same operator (the dietitian involved in this study); segmental and total body composition measurements [25], according to a standardized protocol [24], were taken as follows: First, information (i.e., age, sex, and height) of participants was entered into the device [26]. Next, participants were instructed to stand in a stable, barefoot position. Separate readings for different body regions (i.e., trunk, legs, and arms) were obtained based on an algorithm incorporating impedance, age, and height to estimate total body fat (BF), fat-free mass (FFM), and total body water (TBW) [26]. Participants adhered to all the protocol recommendations for correct BIA measurement, e.g., taking measurements more than three hours (3 h) after waking, urinating before measurements were taken, avoiding food and drink for at least 8 h beforehand, avoiding strenuous exercise, alcohol, and energy drinks for 12 h beforehand, and not having any metal objects or pacemakers.

We considered the following variables:BF = total body fat expressed in kg;BF% (BF as a percentage of the total mass) = (BF ÷ body weight) × 100;Appendicular Lean Mass (ALM) = total lean mass in arms and legs with bone excluded, expressed in kg.

### 2.4. Functional Test

Participants’ gait speed was measured via the four-meter gait-speed test [7], considered physical performance, using international guidelines. The test was performed on flat ground in a room within our clinics, with the distance marked out by tape [7]. Timing was assessed with a chronometer, starting from when the participant began to move until the point at which they had completely crossed the four-meter line [7].

Participants were instructed to ‘walk as quickly as possible’ for four meters (from a moving start). There were two-meter areas in which to accelerate and decelerate on either side of the test distance. Distances (0–2–6–8 m) were marked with tape strips on the floor. The test was performed three times (with a two-minute interval period), and the fastest time to complete the test was used for analysis [7]. The 0.8 m/s cut-point was used to represent a slow walking speed in community-dwelling older people, as defined by the International Academy on Nutrition and Aging (IANA) Task Force [27].

### 2.5. Dietary Practice and Dietary Adequacy (DA) Score

All participants were interviewed by a dietitian and completed a validated version of the Mini Nutritional Assessment (MNA) tool [28]. To this end, and prior to that, they were instructed on food-group servings and portions through food models available in our department.

The dietary items were retrieved from MNA and then categorized to calculate the DA score, all according to the dietary habits of the Lebanese population, as follows [29]:-Number of meals:
Score 0: one meal per day.Score 1: two or three meals per day.

-Protein intake:
Score 0: consumption of no more than one of the three main sources of protein, with the sources defined as follows: one serving of dairy per day; two or more servings of legumes and eggs per week; or meat, fish, or poultry every day.Score 1: consumption of two sources.Score 2: consumption of three sources.

-Fruits and vegetables:
Score 0: fewer than two servings of fruits and vegetables per day.Score 1: two or more servings of fruits and vegetable per day.

-Fluids:
Score 0: less than three-to-five cups per day.Score 1: three-to-five cups per day.


The total DA score was calculated as the sum of the scores for each of the four items, giving a minimum possible score of 0 and a maximum possible score of 5. The DA score was further categorized into a dichotomous variable based on the median score, where a new score of 0 was given to DA scores below the median and a new score of 1 was given to DA scores that were at or above the median.

### 2.6. Statistical Analysis

Means and standard deviations were reported for continuous variables, frequencies, and proportions for categorical variables, respectively. The Student’s *t*-test and chi-squared test for independence were performed to compare means and proportions, respectively. Simple and adjusted logistic regression analyses were performed to assess the association between slow gait speed and low DA while correcting for potential confounders. All tests were considered significant at *p* < 0.05. All statistical analyses were conducted using SPSS v.26 (2019) [30].

## 3. Results

A total of 222 older adults with overweight or obesity were included in this study. Participants were aged 67.7 ± 6.6 years with a mean BMI of 31.9 ± 4.5 kg/m^2^. Almost half of the participants were women (49.1%), most were married (68.0%), had a low education level (82.9%), and were not employed (78.4%) (Table 1). Compared to participants with normal gait speed, those with slow gait speed were older (70.5 ± 6.9 vs. 66.3 ± 5.9 years), more likely to be women (70.1% vs. 37.9%), and less likely to be married (45.5% vs. 80.0%) or employed (89.6% vs. 72.4%) (Table 1). Participants with slower gait speed were also shorter (155.9 ± 9.1 vs. 162.7 ± 8.9 cm), had a higher BF% (34.9 ± 6.3% vs. 31.7 ± 8.0%; *p* = 0.001), and had a lower ALM (20.9 ± 5.1 vs. 23.4 ± 4.9 kg; *p* = 0.001). Weight and BMI did not differ between participants with normal or slow gait speeds (Table 1).

The mean total score and distribution of DA score items by gait speed categories are shown in Table 2. The frequency of consumption for each item did not differ significantly between gait speed categories, although those with lower gait speed were more likely to consume fewer meals and to consume less dairy, legumes, eggs, meat, fish, and poultry. The mean total DA score was significantly higher among those with normal gait speed than among those with slow gait speed (3.37 ± 1.07 vs. 2.99 ± 1.12; *p* = 0.017), reflecting the synergy of the items. Moreover, the participants with slower gait speed were more likely to have a DA score below the median (70.1%) compared to those with normal gait speed (51.7%; X^2^ = 6.999; *p* = 0.017) and more likely to be nutritionally at-risk (22.1% vs. 10.3%; X^2^ = 5.612; *p* = 0.018). When comparing the distribution of the likelihood of having slower gait speed based on DA, those with lower DA were more likely to have slower gate speed (41.9%) than those with higher dietary adequacy (24.7%) (Figure 1).

Simple logistic regression analysis showed that the risk of slow gait speed increased significantly with age (OR = 1.11; 95% CI = 1.05–1.16) or BF% (OR = 1.06; 95% CI = 1.02–1.10). It decreased significantly with ALM (OR = 0.9; 95% CI = 0.85–0.96), in men (OR = 0.26; 95% CI = 0.14–0.47), in those who were married (OR = 0.21; 95% CI = 0.11–0.38), employed (OR = 0.30; 95% CI = 0.13–0.69), or with a DA score above the median DA score (OR = 0.46; 95% CI = 0.25–0.82). However, in the multivariate logistic regression model, only sex, age, and DA were significant determinants. The final model showed that the risk of slow gait speed was 75% lower among those with higher DA scores (OR = 0.25; 95% CI = 0.11–0.53), 93% lower among men (OR = 0.07; 95% CI = 0.01–0.43), and increased by 16% with every year of age (OR = 1.16; 95% CI = 1.09–1.24) (Table 3).

## 4. Discussion

### 4.1. Findings and Concordance with Previous Studies

The current study aimed to assess the prevalence of slow gait speed and its association with dietary practice in a population of community-dwelling older adults with overweight or obesity across a wide age range of 60–85 years old. There were two main findings. First, assessment via the four-meter gait-speed test showed that the prevalence of slow gait speed in our population was estimated at nearly 35%. To the best of our knowledge, this is the first time that the prevalence of slow gait speed has been reported among Lebanese community-dwelling older adults; therefore, it is not possible to compare this finding with previous studies conducted in Lebanon. A previous large study (n = 1327), conducted in Spain with an urban population aged 65 and above recruited from primary care, found a similar prevalence of around 40% [31]. However, our finding stands in contrast to a prevalence of about 50% estimated in the United States (US) [32]. This difference might be attributed to a difference in dietary practices between Lebanese and Spanish populations (i.e., Mediterranean) [31] and an American population (i.e., Western) [32,33].

Second, we found that participants with reduced gait speed had a lower DA mean score, falling below the median, than those with normal gait speed. Participants with a slower gait speed were also more likely to be nutritionally at-risk, and the risk of having a slow gait speed was 75% lower among those with a higher DA score. This finding suggests that having a higher DA score might be associated with increased gait speed. Our findings align with previous literature; in particular, a study conducted in the US, investigating the association between overall diet quality, measured by the US Department of Agriculture’s Healthy Eating Index-2005 (HEI-2005), and physical performance, measured by gait speed, found that major adherence to overall dietary recommendations (i.e., higher HEI-2005 scores) was associated with better physical performance among older adults [34]. Similarly, in line with our finding and within a nationwide population-based sample of older adults, Mendes and colleagues showed that individuals with overweight or obesity and nutritional risk (a score < 12 points assessed with the MNA-SF) had slow gait speed [23]. However, Mendes and colleagues only analyzed dietary patterns according to the total MNA-SF score without going through the participants’ dietary habits. Despite the fact that we have used the same tool (i.e., MNA-SF), we have established a specific sub-scoring system that is able to assess the DA of an individual in relation to Lebanese habits [23] associated with a normal gait speed, namely reporting the intake of: at least two meals per day; at least two servings of fruits and vegetables per day; three to five cups of fluids/day; and at least two protein sources per day. We think that is more useful since it improves the evaluation of the dietary practice of an individual related to slow/normal gait speed.

Our second finding of an association between low DA and slower gait speed should be interpreted with extreme caution because of the cross-sectional nature of our study. It seems that impairment in the four-meter gait-speed test could be used to screen individuals at nutritional risk from low DA. Our findings raise an interesting question as to whether higher DA may improve gait speed, but longitudinal interventions are needed to determine a cause-effect relationship between diet and physical performance in this population (i.e., those with overweight/obesity).

### 4.2. Potential Clinical Implications

There are several clinical implications of our study. First, there is a high prevalence of slower gait speed among community-dwelling older adults with overweight or obesity. Health professionals should be made aware of this issue. Given the importance of this variable in predicting clinical outcomes and mortality, routine assessment by all health professionals dealing with this population in easily accessible settings such as primary care (i.e., general practice) is recommended. Second, in line with the first implication, and since slow gait speed is associated with low DA, health professionals should refer patients with slow gait speed to a specialist (i.e., dietitians or nutritionists) for a detailed investigation of their nutritional practice to ensure that it meets recommendations and guidelines. If specialist referral is not possible because of limited resources, health professionals could consider assessing nutritional risk through questionnaires that have been validated for this purpose in this population. Third, adequate nutrition may potentially improve gait speed, but this should be confirmed through longitudinal investigation to see if adequate dietary patterns can determine improvement in gait speed over a period of time.

### 4.3. Strengths and Limitations

This study has some strengths. To the best of our knowledge, it is the first to investigate the association between gait speed and DA among community-dwelling older adults with overweight or obesity in Lebanon, a country within the Middle East and North Africa (MENA) region particularly affected by overweight and obesity [35]. It also has clinical applicability and implications: as there is a high prevalence of slower gait speed among community-dwelling older people with overweight or obesity, associated with low DA, and given that these variables are important predictors of clinical outcomes and mortality, the issue should be openly discussed with older adults with overweight or obesity to highlight the necessity of regular monitoring of gait speed and DA.

This study has several limitations. First, we included only community-dwelling older adults with overweight and obesity; therefore, our findings lack external validity and cannot be generalized to older adults in other settings (e.g., those living in retirement homes, those who have been hospitalized, etc.) [36]. Second, the non-controlled cross-sectional design cannot elicit information regarding comparison between our population and those within the normal weight status, nor information on causality [37], and our findings need to be replicated in a longitudinal-controlled study to detect the directional, cause–effect relationship between low DA and slow gait speed, as well as any potential difference from findings in relation to the normal weight population. Third, while physical exercise levels are known to have an impact on gait speed and well-structured physical activity seems to produce a faster gait speed [38], our study lacked an objective assessment of physical exercise levels. Fourth, this study lacked an accurate macro- and micro-nutritional assessment of calories and micronutrients; such an assessment would allow us to determine with more precision the specific nutritional component associated with slow gait speed, which may be more helpful in implementing more personalized nutritional interventions. In line with this limitation, the main outcome related to DA was dichotomized as “normal DA” or “low DA”; therefore, we were not in a position to determine the continuous association between DA and gait speed scores [39].

Fifth, we assessed body composition with the BIA, which is considered an approximate measurement compared to the gold standard technique of Dual-Energy X-ray Absorptiometry (DEXA). Nevertheless, the multi-frequency BIA we used has been found to be reasonably accurate in individuals with overweight or obesity when compared to DEXA [40]. Finally, the lack of biochemical testing means that we were unable to understand exactly the nature of the interaction between low DA and slow gait speed [41]. Future studies will be required to identify underlying molecular mechanisms and their contributions.

### 4.4. New Directions for Future Research

Our findings highlight the strong association between dietary practice and gait speed, but further work is required to draw firm conclusions. Primarily, there is a need to determine if adequate dietary practice is effectively an independent factor that can improve gait speed. Randomized controlled trials of longitudinal nutritional interventions that adjust for confounders such as body mass (i.e., BMI, BF, and lean mass) or other factors (e.g., morbidities, lifestyle behaviors, etc.) should be conducted to determine with certainty the exact role of nutrition. To support such trials, precise nutritional indications are needed, such as specific diets with precise bromatological composition and dietary combinations and specific micronutritional information, to support the examination of supplementation. Another area of interest would be to explore further the impact of dietary practice on gait speed to understand what happens after the improvement of the latter in terms of hard, long-term outcomes (e.g., mortality, etc.).

## 5. Conclusions

Overweight and obesity are common health conditions with increasing global prevalence across the lifespan of individuals, including among older people. Both overweight and obesity seem to contribute substantially to the burden of many chronic health conditions; excess body weight is considered a common denominator of overweight and obesity and has been found to be more correlated with a higher risk of cardio-metabolic diseases, osteoarthritis, some cancers, and an unavoidable increase in mortality risk.

Another relevant health issue for older people is frailty, defined as a biological condition that produces a poor resolution of several physiological systems to maintain homoeostasis after a low-power stressor event, in particular, impairment of physical performance linked to a higher risk of falls, disability, hospitalization, and mortality. In this context, our study provides evidence of a significant association between low DA and slow gait speed among community-dwelling older adults of both sexes with overweight or obesity in Lebanon. Slow gait speed is an important health issue with several complications that might be prevented and improved by monitoring and managing DA. Future studies are needed to verify the cause–effect relationship between nutrition and physical performance and, once confirmed, identify specific nutritional interventions that can improve gait speed.

## Figures and Tables

**Figure 1 diseases-12-00054-f001:**
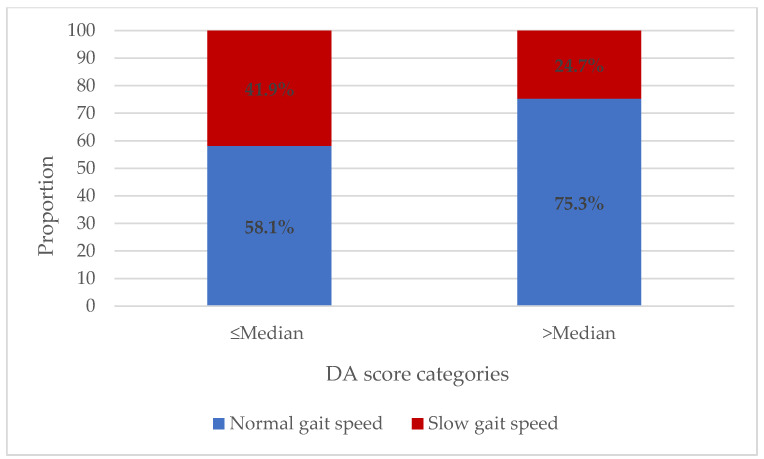
Distribution of gait speed according to median DA score.

**Table 1 diseases-12-00054-t001:** Anthropometric, body composition, and socio-demographic characteristics of the study participants.

	Total (n = 222)	Slow Gait Speed (≤0.8 m/s) (n = 77)	Normal Gait Speed (>0.8 m/s) n = 145)	Significance
Age (years)	67.6 (6.6)	70.5 (6.9)	66.3 (5.9)	*p* < 0.0001
Sex				X^2^ = 20.86; *p* < 0.0001
Male	113 (50.9)	23 (29.9)	90 (62.1)	
Female	109 (49.1)	54 (70.1)	55 (37.9)	
Marital status				X^2^ = 27.59; *p* < 0.0001
Not married	71 (32.0)	42 (54.5)	29 (20.0)	
Married	151 (68.0)	35 (45.5)	116(80.0)	
Level of education				X^2^ = 0.666; *p* = 0.414
Lower education	184 (82.9)	66 (85.7)	118 (81.4)	
Higher education	38 (17.1)	11 (14.3)	27 (18.6)	
Employment				X^2^ = 8.776; *p* = 0.003
Not employed	174 (78.4)	69 (89.6)	105 (72.4)	
Employed	48 (21.6)	8 (10.4)	40(27.6)	
Weight (kg)	82.2 (14.6)	79.6 (15.8)	83.6 (13.9)	*p* = 0.061
Height (cm)	160.3 (9.6)	155.9 (9.1)	162.7 (8.9)	*p* < 0.0001
BMI (kg/m^2^)	31.9 (4.5)	32.6 (4.8)	31.6 (4.3)	*p* = 0.114
BF (kg)	27.2 (8.7)	28.0 (8.1)	26.7 (8.9)	*p* = 0.273
BF (%)	32.8 (7.6)	34.9 (6.3)	31.7 (8.0)	*p* = 0.001
ALM (kg)	22.5 (5.1)	21.0 (5.1)	23.4 (4.9)	*p* = 0.001

Values are expressed in (%) for categorical variables and mean (SD) for continuous variables. BMI = body mass index; BF = body fat; BF% = body fat percentage; and ALM = appendicular lean mass.

**Table 2 diseases-12-00054-t002:** Distribution of dietary items and DA scores according to gait speed.

	Total (n = 222)	Slow Gait Speed (≤0.8 m/s) (n = 77)	Normal Gait Speed (>0.8 m/s) (n = 145)	Significance
Meals per day				X^2^ = 2.470; *p* = 0.116
1 meal	15 (6.8)	8 (10.4)	7 (4.8)	
≥2 meals	207 (93.2)	69(89.6)	138 (95.2)	
Protein intake per day				X^2^ = 4.671; *p* = 0.097
At least 1 source	87 (39.2)	36 (46.8)	51 (35.2)	
2 sources	94 (42.3)	32 (41.6)	62 (42.8)	
3 sources	41 (18.5)	9 (11.7)	32 (22.1)	
Fruit and vegetable intake per day				X^2^ = 2.640; *p* = 0.104
<2 servings a day	71 (32.0)	30 (39.0)	41 (28.3)	
≥2 servings a day	151 (68.0)	47 (61.0)	104 (71.7)	
Fluid intake per day				X^2^ = 0.004; *p* = 0.950
<3 cups	37(16.7)	13(16.9)	24 (16.6)	
3–5 cups	185 (83.3)	64 (83.1)	121 (83.4)	
Total DA score	3.24 (1.10)	2.99 (1.12)	3.37 (1.07)	*p* = 0.017
				X^2^ = 6.999; *p* = 0.008
<Median	129 (58.1)	54 (70.1)	75 (51.7)	
≥Median	93 (41.9)	23 (29.9)	70 (48.3)	
Nutritional status				X^2^ = 5.612; *p* = 0.018
Normal	190 (85.6)	60 (77.9)	130 (89.7)	
At risk (Low DA)	32 (14.4)	17 (22.1)	15 (10.3)	

Values are expressed in (%) for categorical variables and mean (SD) for continuous variables. Low DA = low dietary adequacy that falls below the median.

**Table 3 diseases-12-00054-t003:** Association between DA and gait speed before (bivariate analysis) and after adjustment (multivariable regression) for socio-demographic variables.

	Bivariate Analysis		Multivariable Regression	
Variables	OR	95%CI	OR	95%CI
Age (years)	1.11	1.05–1.16	1.16	1.09–1.24
Sex				
Female	1.00			
Male	0.26	0.14–0.47	0.07	0.01–0.43
BMI (kg/m^2^)	1.05	0.99–1.12	1.16	0.97–1.36
BF (%)	1.06	1.02–1.10	0.90	0.80–1.00
ALM (kg)	0.90	0.85–0.96	1.07	0.90–1.28
Marital status				
Not married	1.00			
Married	0.21	0.11–0.38	0.53	0.25–1.14
Employment				
Not employed				
Employed	0.30	0.13–0.69	1.03	0.37–2.86
DA score				
<Median	1.00			
≥Median	0.46	0.25–0.82	0.25	0.11–0.53

BMI = body mass index; BF% = body fat percentage; ALM = appendicular lean mass; and DA = dietary adequacy.

## Data Availability

Data are available from the corresponding author on reasonable request.

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
