# Peer review of "Association between Dietary Practice and Gait Speed in Community-Dwelling Older Adults with Overweight and Obesity: A Cross-Sectional Study"

_diseases, 2024, doi:10.3390/diseases12030054_

Round 1
Reviewer 1 Report
Comments and Suggestions for Authors
It was interesting to read the manuscript by Itani et al titled ‘Association Between Dietary Practice and Gait Speed in Com- 2 munity-Dwelling Older Adults with Overweight and Obesity: 3 A Prospective Cross-sectional Study ‘on the effect of dietary adequacy and gait speed. In general, the paper is well written.
The paper will benefit by introducing more on the concept of dietary adequacy and how it is defined. How is nutritional status or assessing malnutrition different from dietary adequacy. Do the authors have previous data on the gait speed and dietary adequacy in normal weight individuals of the same age group. Why was the normal weight group not included along with the overweight and obese group?
Were the questionnaires used for the assessments validated? In line 263 the authors mentioned ‘but we established a specific sub-scoring system’ it is not clear what this sub-scoring system is. It will be beneficial if this is explained in detail and how this improves the evaluation of the patients.
The authors have mentioned clearly the limitations of the study. Few of these limitations could have been easily overcome, why were these not considered when the study was conducted.
Comments on the Quality of English LanguageThe english language is fine, but needs minor corrections.
Author Response
Reviewer 1#
It was interesting to read the manuscript by Itani et al titled ‘Association Between Dietary Practice and Gait Speed in Community-Dwelling Older Adults with Overweight and Obesity: A Prospective Cross-sectional Study ‘on the effect of dietary adequacy and gait speed. In general, the paper is well written.
The paper will benefit by introducing more on the concept of dietary adequacy and how it is defined. How is nutritional status or assessing malnutrition different from dietary adequacy. Do the authors have previous data on the gait speed and dietary adequacy in normal weight individuals of the same age group. Why was the normal weight group not included along with the overweight and obese group?
Response: we introduced more on the concept of dietary adequacy and how it is defined, and its difference from malnutrition in the Introduction section (Page 2; paragraph 3).
No data was available on normal-weight population as control group. We added this as a limitation in the Discussion section (Page 9; paragraph 3).
Were the questionnaires used for the assessments validated? In line 263 the authors mentioned ‘but we established a specific sub-scoring system’ it is not clear what this sub-scoring system is. It will be beneficial if this is explained in detail and how this improves the evaluation of the patients. Response: we thank the reviewer for the comment. Now we clarified as suggested. Firstly in the Methods and Materials section how the DA scoring system was retrieved from the MNA validated questionnaire (Page 3; paragraph 1 from the “Dietary practice and dietary adequacy (DA) score” subsection). Moreover in the Discussion section we highlighted its importance for patients (Page 8; paragraph 2).
The authors have mentioned clearly the limitations of the study. Few of these limitations could have been easily overcome, why were these not considered when the study was conducted. Response: we thank the reviewer, and we will promise that we will do our best in our future studies.
Reviewer 2 Report
Comments and Suggestions for Authors
Thank you for the opportunity to review the paper titled “Association Between Dietary Practice and Gait Speed in Community-Dwelling Older Adults with Overweight and Obesity: 3 A Prospective Cross-sectional Study ”. The paper was enjoyable to read, was generally well written, and has sufficient literature to support the hypothesis. My comments, therefore, are very few.
-Line 138, Methods Section on protein intake: A score of 0 is defined as ‘at least 1’ of the sources of protein. However, ‘at least’ one means there could be more, which would then be the same as a score of 1 or 2. The score of Zero should probably be ‘no more than 1’, or ‘0-1 sources’.
I appreciate the caution of interpreting the result due to the cross sectional design. Additionally, there is no clarity on the direction of the relationship between gait speed and DA, which seems important on this topic. Third, gait speed is a function of leg strength and power, neither of which were measured here, but may be able to shed more light on the inter-relationships. And finally, was there an opportunity to further separate the sample into a third category of ‘high’ gait speed, to see if the relationships with DA continued?
Author Response
Reviewer 2#
Thank you for the opportunity to review the paper titled “Association Between Dietary Practice and Gait Speed in Community-Dwelling Older Adults with Overweight and Obesity: A Prospective Cross-sectional Study ”. The paper was enjoyable to read, was generally well written, and has sufficient literature to support the hypothesis. My comments, therefore, are very few.
-Line 138, Methods Section on protein intake: A score of 0 is defined as ‘at least 1’ of the sources of protein. However, ‘at least’ one means there could be more, which would then be the same as a score of 1 or 2. The score of Zero should probably be ‘no more than 1’, or ‘0-1 sources’. Response: Totally agree, corrected as suggested (Page 4; paragraph 1 of Protein Intake subsection).
I appreciate the caution of interpreting the result due to the cross sectional design. Additionally, there is no clarity on the direction of the relationship between gait speed and DA, which seems important on this topic. Third, gait speed is a function of leg strength and power, neither of which were measured here, but may be able to shed more light on the inter-relationships. And finally, was there an opportunity to further separate the sample into a third category of ‘high’ gait speed, to see if the relationships with DA continued? Response: our main outcome regard the DA was a dichotomized variable: “normal DA” and “Low DA” one it is above or below the median, since it cannot be expressed as a continuous variable, therefore we are not in the position to assess the continuous association low, medium, high gait speeds and DA. However add this point to the limitations of the study (Page 9; paragraph 3).
Reviewer 3 Report
Comments and Suggestions for Authors
General
The manuscript of Itani et al. focuses on dietary habits and gait speed among Lebanese seniors. The article is generally well-writen, however it requires some specific changes which are marked below:
Introduction
-In my opinion, the whole introduction needs better coherence. Now it seems that there are a few parts of the introduction that are poorly associated with each other (at the beginning you mention the age-related decline in adults, then about the gait speed test, then the problem of being overweight/obese, and then once again about gait speed test. I think it is necessary to expand each of these issues and make them more related to each other.
-Rows 37-39 – in my opinion they would better fit the Materials and Methods section.
-“However, the association between nutritional status and gait speed remains unclear” – it is not true, as I have already found a few articles about such an association. It is crucial to cite and describe them, as well as indicate what differentiates your manuscript from the ones (what is the novelty?):
https://www.ncbi.nlm.nih.gov/pmc/articles/PMC5844922/
https://pubmed.ncbi.nlm.nih.gov/37950363/
Materials and Methods
-Row 118 – are there any reference values regarding the gait speed test for seniors? In the Table 1 I can see a category for slow and normal gait speed.
-Did you provide any additional information for participants while they were answering to Mini Nutritional Assessment, regarding the serving size of specific products? Participants may not have had proper knowledge of for example, what the serving of fruit or vegetable is.
Results
-Table 2 – what do you mean by “low DA” – below the median? You should clarify it.
References
-I would suggest to add another references in order to enrich the Discussion section.
Author Response
Reviewer 3#
General
The manuscript of Itani et al. focuses on dietary habits and gait speed among Lebanese seniors. The article is generally well written, however it requires some specific changes, which are marked below:
Introduction
-In my opinion, the whole introduction needs better coherence. Now it seems that there are a few parts of the introduction that are poorly associated with each other (at the beginning you mention the age-related decline in adults, then about the gait speed test, then the problem of being overweight/obese, and then once again about gait speed test. I think it is necessary to expand each of these issues and make them more related to each other. Response: we did our best to improve the whole Introduction section as suggested (Page 1 and 2).
- Rows 37-39 – in my opinion they would better fit the Materials and Methods section. Response: we rephrase the statement to appear more suitable for the Introduction section (Page; paragraph). Moreover the description of the four-meter gait speed has been also improved in the Materials and Methods section (Page 1; paragraph 1 of the Introduction section).
-“However, the association between nutritional status and gait speed remains unclear” – it is not true, as I have already found a few articles about such an association. It is crucial to cite and describe them, as well as indicate what differentiates your manuscript from the ones (what is the novelty?):
https://www.ncbi.nlm.nih.gov/pmc/articles/PMC5844922/ [23]
https://pubmed.ncbi.nlm.nih.gov/37950363/ [9]
Response: The statement has been improved as follows:
Recent literature has also highlighted an association between slow gait speed, overweight and obesity in a nationwide population-based sample of older adults [23]; and in this study, an association between nutritional status and gait speed was found and reported [23], however this interesting aspect has been widely examined by many studies [23] for this reason remains not fully understood (Page 2; paragraph 4).
In fact the reference [23] is exactly the reference suggested by the reviewer, was already mentioned in the introduction and widely discussed and compared with our study, in the Discussion section (Page 8; paragraph 2).
On the other hand, the second suggested reference is not related to our population and topic of study, since we were focused on community-dwelling older adults with overweight and obesity mainly not affected by major chronic diseases. However, we added it in the Introduction section as reference n° [9] while taking about the gait speed test and diseases (Page 1; paragraph 1).
Materials and Methods
-Row 118 – are there any reference values regarding the gait speed test for seniors? In the Table 1 I can see a category for slow and normal gait speed. Response: yes, and now is reported in the Method section under the subsection “Functional test” (Page 3; second paragraph of this subsection) with a suitable reference (n° [27]).
-Did you provide any additional information for participants while they were answering to Mini Nutritional Assessment, regarding the serving size of specific products? Participants may not have had proper knowledge of for example, what the serving of fruit or vegetable is. Response: yes, this has been done, and now clearly mentioned in the Method section under the subsection “Dietary practice and dietary adequacy (DA) score” (Page 3; first paragraph of this subsection).
Results
-Table 2 – what do you mean by “low DA” – below the median? You should clarify it. Response: Yes it is below the median, now this is clarified in the Table 2
References
-I would suggest to add another references in order to enrich the Discussion section. Response: all in all we added 5 references in the manuscript some are in the Discussion section (n° [9][22][27][39][41].
Reviewer 4 Report
Comments and Suggestions for Authors
It is very interesting study examining the association between the Dietary Adequacy score system and walking speed. There are some points needed to be revised.
-With regard to the title, the study appears to be a cross-sectional study. Therefore, titles containing the word Prospective may not accurately reflect the study design.
-With regard to ALM, the Skelta muscle mass index, which is ALM divided by the square of height, is usually used to assess muscle mass, and it is also used in many diagnostic criteria for sarcopenia.
Author Response
Reviewer 4 #
It is very interesting study examining the association between the Dietary Adequacy score system and walking speed. There are some points needed to be revised.
-With regard to the title, the study appears to be a cross-sectional study. Therefore, titles containing the word Prospective may not accurately reflect the study design. Response: The word prospective was removed from the Title and Method section (Page 2; paragraph 5) as suggested.
-With regard to ALM, the Skelta muscle mass index, which is ALM divided by the square of height, is usually used to assess muscle mass, and it is also used in many diagnostic criteria for sarcopenia. Response: ALM is the general expression of the appendicular muscle mass, and we aware about indices when divided by height or some time by weight or BMI as expression of sarcopenia. However we preferred not to include it, since we believe that it is out of scope of our study. We thank the reviewer and will take this into consideration in future studies.